# Enhancement of Classifier Performance with Adam and RanAdam Hyper-Parameter Tuning for Lung Cancer Detection from Microarray Data—In Pursuit of Precision

**DOI:** 10.3390/bioengineering11040314

**Published:** 2024-03-26

**Authors:** Karthika M S, Harikumar Rajaguru, Ajin R. Nair

**Affiliations:** 1Department of Information Technology, Bannari Amman Institute of Technology, Sathyamangalam 638401, India; karthikams@bitsathy.ac.in; 2Department of Electronics and Communication Engineering, Bannari Amman Institute of Technology, Sathyamangalam 638401, India; ajinrnair@bitsathy.ac.in

**Keywords:** lung cancer detection, MAGE data, DimRe, cancer classification, Adam and RanAdam tuning, FFT, mixture model

## Abstract

Microarray gene expression analysis is a powerful technique used in cancer classification and research to identify and understand gene expression patterns that can differentiate between different cancer types, subtypes, and stages. However, microarray databases are highly redundant, inherently nonlinear, and noisy. Therefore, extracting meaningful information from such a huge database is a challenging one. The paper adopts the Fast Fourier Transform (FFT) and Mixture Model (MM) for dimensionality reduction and utilises the Dragonfly optimisation algorithm as the feature selection technique. The classifiers employed in this research are Nonlinear Regression, Naïve Bayes, Decision Tree, Random Forest and SVM (RBF). The classifiers’ performances are analysed with and without feature selection methods. Finally, Adaptive Moment Estimation (Adam) and Random Adaptive Moment Estimation (RanAdam) hyper-parameter tuning techniques are used as improvisation techniques for classifiers. The SVM (RBF) classifier with the Fast Fourier Transform Dimensionality Reduction method and Dragonfly feature selection achieved the highest accuracy of 98.343% with RanAdam hyper-parameter tuning compared to other classifiers.

## 1. Introduction

Cancer is a major threat and health concern worldwide. It is a medical condition characterised by the unregulated growth of abnormal cells. Different types of cancers occur in virtually any tissue or organ in the body as mentioned in Egeblad et al. [1]. Among the different cancer types, lung cancer is one of the leading causes of cancer-related deaths worldwide, as reported by Dela et al. [2]. It is considered the most dangerous type of cancer due to several factors, such as late diagnosis, rapid spread, limited treatment options, poor survival rate, etc., as mentioned by Schabath et al. [3]. Lung cancer begins in the cells of the lungs and is primarily caused by smoking, as indicated by Alaoui et al. [4]. Additional risk factors for lung cancer include a familial background of the disease and prior chest radiation therapy, exposure to second-hand smoke and occupational exposure to certain hazardous substances like asbestos, arsenic, diesel exhaust, and chromium as pointed out in Mustafa et al. [5]. The survival rates of lung cancer are highly dependent on the prognosis of cancer at early stages. Next, we discuss some of the related research literature associated with lung cancer prognosis.

### Review of Previous Work

As suggested in Dela et al. [2], the early detection and identification of lung cancer tissues will increase the survival rate. Diagnosing lung cancer involves a combination of medical history assessment, physical examination, and clinical techniques such as Chest X-rays, Computer Tomography (CT) scans, Sputum Cytology, Bronchoscopy, Positron Emission Tomography (PET) Scans, etc., to effectively detect cancer tissue presence in the human body. CT scans produce detailed cross-sectional images of the lungs by utilising X-ray images captured from various angles. CT scans can provide more precise information about lung nodules or a tumour’s size, shape, and location, Causey et al. [6]. Sputum cytology involves examining a sample of mucus coughed up from the lungs under a microscope. It is mainly used to detect lung cancer in individuals with coughing, chest pain, or shortness of breath, Mukae et al. [7]. The sensitivity and specificity of these clinical procedures can be used to explain their major limitations. Chest X-rays have relatively low sensitivity, mainly for detecting cancer cells in the early detection stages, as indicated by Konstantina Kourou et al. [8]. Also, high radiation exposure often makes CT cumbersome. Sputum Cytology also faces issues like low sensitivity, particularly in the early stages of cancer, and dependency on the presence of cancer cells in the collected sputum. Bronchoscopy collects small lesions from peripheral lung areas and may contain potential false negatives, Leong et al. [9]. Like the Bronchoscopy technique, PET Scans analyse small lesions to distinguish benign and malignant abnormalities but have limited spatial resolution, Visser et al. [10].

Furthermore, invasive procedures such as Bronchoscopy and Sputum Cytology carry the potential risk of severe complications, including bleeding, pain, and infection, and it is only possible to detect malignant cells. So, there are inherent risks invested in collecting tissue samples for the above methods as mentioned in Rivera et al. [11]. Hence, these methods are suggested when an oncologist witnesses significant and solid observations in the early stages of lung cancer. 

For the above concern, as suggested in Lubitz et al. [12], the MicroArray Gene Expression (MAGE) data analysis is often preferred due to its use of minimally invasive methods such as fine needle biopsies and blood tests for sample collection. The microarray method comforts and lowers the overall risk profile, making molecular analysis a safer alternative for obtaining diagnostic information, Dhaun et al. [13]. The MAGE data analysis provides a comprehensive molecular profile of the tumour, allowing for a detailed understanding of the genetic alterations associated with lung cancer. This way, the microarray data analysis is unique compared to bronchoscopy and sputum cytology methods, which may only detect malignant cells. The microarray method can unveil specific genetic compositions and its mutations, MAGE patterns, and molecular signatures indicating lung cancer’s possibility. Thus, the microarray method aids accurate diagnosis and envisages personalised treatment strategies based on unique genetic characteristics.

MAGE data is typically a high-dimensional dataset containing measurements of thousands of gene expression levels, as discussed in Nguyen et al. [14]. Data analysis is difficult due to the large number of features, which makes it cumbersome to visualise the relationships between the genes. This problem is often regarded as the curse of dimensionality, as mentioned in Saheed et al. [15]. In [15], the authors have suggested dimensionality reduction (DimRe) as an effective tool to improve the classification performance of the Machine Learning (ML) classification model for MAGE data. The DimRe process aims to decrease the number of features in a dataset while retaining the crucial information. The DimRe methods facilitate the identification of patterns and relationships within the data at subspace, ultimately enhancing the effectiveness of ML algorithms. Further, Feature Selection (FS), as performed by Jager et al. [16], refines the features obtained after DimRe to improve the classification performance. 

In De Souza et al. [17], Principal Component Analysis (PCA) was suggested as one of the methods for DimRe in lung cancer MAGE data sets. However, PCA does not capture the inherent nonlinear relationships in the MAGE data. A t-distributed Stochastic Neighbor Embedding (t-SNE) is proposed as a DimRe method by Rafique et al. [18]. However, t-SNE is sensitive to the choice of hyper-parameters, and different runs may yield different results. Inamura et al. [19] utilised Non-negative Matrix Factorisation (NMF) as a DimRe for MAGE data. The NMF is very sensitive to initialisation conditions, leading to potential results variability, especially when applied to microarray data. Sparse Principal Component Analysis (Sparse PCA) was proposed by Hsu et al. [20] for the processing of lung cancer microarray datasets. The major challenge in Sparse PCA that impacts the overall results is selecting an appropriate sparsity parameter. Mollaee et al. [21] utilised Independent Component Analysis (ICA) to reduce MAGE data’s dimensionality. However, the ICA assumes statistical independence in the data, which may only hold well in complex biological datasets like the MAGE data. For DimRe, Chen et al. [22] proposed LASSO (Least Absolute Shrinkage and Selection Operator) for MAGE analysis of Adenocarcinoma and lung squamous cell carcinoma medical conditions. The LASSO requires careful tuning of regularisation parameters and selecting an optimal parameter might be data dependent. The use of a Genetic Algorithm (GA) FS and manifold learning technique is implemented by Wang et al. [23] for cancer classification using microarray data. This Isomap (Isometric Mapping) technique is computationally expensive and sensitive to noise in the data. So, absolute noise removal is essential for a properly working Isomap in MAGE data analysis. The methods like Locally Linear Embedding (LLE) are investigated by Lee et al. [24] for DimRe in MAGE data. LLE is sensitive to the choice of neighbours, and results may vary with different parameter settings. The Fast Fourier Transform (FFT) technique is utilised for the DimRe of DNA methylation data by Raweh et al. [25]. FFT-aided classification reported accurate results with reduced training time. FFT enables the fast computation of frequency components and reduces the training time. Also, the Frequency Domain Interpretation enhances the classification accuracy by revealing hidden periodic and cyclic patterns in the data, providing insights not easily captured using methods like PCA or t-SNE. Otoom et al. [26] utilised Mixture Model (MM) analysis for the DimRe of breast cancer microarray data. The MM method’s DimRe reported an enhanced classification performance for the ML classifiers. The MM is a probabilistic framework that allows a more nuanced understanding of uncertainty and variability in the microarray dataset. The MM also shows cluster interpretability that can naturally represent clusters within the data, interpreting distinct subgroups of MAGE profiles easier. For the above advantages in this research, we adopt FFT and MM as DimRe methods for the lung cancer microarray data.

After DimRe, the reduced data containing distinct and relevant features are subjected to classification. Orsenigo et al. [27] utilised nonlinear manifold techniques for various cancer microarray data classification. The nonlinear manifold technique reported 81% classification accuracy for the lung cancer microarray data. Independent component analysis with naïve Bayes classification attained 83% accuracy for lung microarray datasets, as reported by Fan et al. [28]. Chen et al. [29] used a combination of particle swarm optimisation and C4.5 decision tree classification for cancer classification from MAGE data that reported an 87% accuracy. Díaz et al. [30] achieved a minimum Out-of-bag (OOB) error rate between 0.1% and 0.2% for Random Forest-based classification with exhaustive evaluation (large tree size). Support Vector Machine (SVM) and Radial Basis Function (RBF) classification is applied by Azzawi et al. [31] for the cancer classification with microarray data, reporting 90% classification accuracy. Given the above research, we choose Nonlinear Regression (NR), Naive Bayesian (NB), Random Forest (RF), Decision Tree (DT) and SVM (RBF) as the classification methods for lung cancer classification from microarray data. However, these classification methods need further enhancement to improve overall classification performance.

One of the significant techniques to improve a classifier’s overall classification outcome is through optimising the parameters associated with the classifier methodology. As sermonised in Kotsiantis et al. [32], the parameters are internal coefficients or weights learned during the training phase of the classifier. The parameters adapt the model’s decision boundaries to represent what it has learned from the input data. Traditionally, a fixed learning rate for parameter updates is chosen in classifiers. However, as mentioned by Ioannou et al. [33], adjusting this rate can impact convergence speed and prevent overfitting or underfitting. Therefore, a hyper-parameter tuning method can control how the classifier model learns from the data. It can optimise the learning rates, regularisation strength, kernel parameters, etc., thereby significantly boosting the classification accuracy and overall performance of the classifier. Hyper-parameter tuning helps to balance the classifier model in terms of complexity and flexibility. In this way, the hyper-parameter tuning can improve the memory of training data, preventing overfitting, and unveiling unseen data and patterns, preventing underfitting.

The grid search is used for parameter tuning by Alrefai et al. [34] to improve the classification performance of microarray data. The Bayesian optimisation is employed by Quitadadmo et al. [35] for microarray data. It is more computationally efficient than Grid Search but may not always outperform Random Search. A momentum back propagation is implemented as parameter tuning for cancer detection from microarray data in Wisesty et al. [36], who reported 94% accuracy in lung cancer classification. Rakshitha et al. [37] used RMSprop (Root Mean Square Propagation) as a tuning technique for classifying and predicting ovary cysts and reported 89% accuracy. Adaptive Moment Estimation (Adam) combines ideas from both momentum optimisation and RMSprop. This optimiser can adjust learning rates based on gradients, offering faster convergence for MAGE data. Sena et al. [38] utilised Adam for ECG classification using Convolutional Neural Networks (CNNs). Random Adaptive Moment Estimation (RanAdam) is an extension of the Adam optimisation algorithm with the addition of randomisation. So, RanAdam anticipates further improvement in the tuning capability compared to the Adam hyper-parameter tuning method. So, based on all the above observations in the literature, this research considers both Adam and RanAdam hyper-parameter tuning methods for improving the classification performance of the ML classifiers.

## 2. Materials and Methods

The well-known Gordon MAGE dataset [39] that contains malignant pleural Mesothelioma and Adenocarcinoma is used for this research. The dataset contains gene expressions useful in lung cancer classification and aid in cancer prognosis at a much earlier stage. 

The overall methodology adopted in this research work consists of three approaches. In all three approaches, DimRe is performed as a first step. The DimRe converts higher dimensional MAGE data into lesser dimensional data, retaining the unseen patterns and significant information. The first approach classifies the data after DimRe using ML classifiers into Adeno and Meso classes. The evaluation of the classifier’s performance involves various performance metrics. The second approach utilises FS methods after reducing dimensionality to remove redundant or noisy information. These relevant features are subjected to classification, and their performance is evaluated. In the third approach, after performing FS, the Adam and RanAdam hyper-parameter tuning is incorporated into the classifiers to optimise the overall performance of the classifiers. The above approaches employed in this research are abridged in Figure 1.

### 2.1. Details about the Dataset

The Gordon dataset [39] comprises two distinct classes: Adenocarcinoma and Mesothelioma. The dataset consists of 181 tissue samples (12,533 × 181), with 150 samples of Adenocarcinoma (12,533 × 150) and 31 of malignant Mesothelioma (12,533 × 31). Here, 12,533 characterise each tissue sample. The total number of rows in the matrix is 12,534, including the last row for class labels: ADCA for Adenocarcinoma and MPM for malignant Mesothelioma samples. The number of patient data for ADCA and MPM are different; there is a data imbalance. The MAGE dataset is built on original surgical specimens from patients with microarray experiments. The method is independent of the platform employed for data acquisition and does not need an integration of the method of transcription to translation for selected genes. These reasons make MAGE ratios a useful method for training and evaluating algorithms for lung cancer classification.

### 2.2. Dimensionality Reduction (DimRe)

As previously stated, the microarray data is inherent with the curse of dimensionality, leading to significant Computational Complexity (CC) and reduced model generalisation. DimRe techniques become imperative here, as they mitigate the risk of overfitting, enhance interpretability, and facilitate more efficient analysis by extracting meaningful patterns from the high-dimensional data. The volume of data is reduced while preserving essential information. So, the method improves accuracy and scalability in analysing MAGE data. The methodology for DimRe using the Mixture Model and the Fast Fourier Transform algorithm is discussed in the next section.

#### 2.2.1. Mixture Model for DimRe

In the MM methodology, each gene’s expression pattern is considered a univariate distribution, as indicated by Liu et al. [40]. So, one or two Gaussian distributions will be fitted for each distribution and then unequally sub-distributed to fractions and variances. The maximisation of Bayesian Information Criteria is the base for selecting the Gaussian fitted distribution. It is expressed as follows:(1)$$B_{A,N}=x\log\;(n)-2\;log(H_{A,N}\left(y|\theta \right))$$
(2)$$B=y^{2}+xln(n)$$
here, model A contains N components with a maximum likelihood function of $\;H_{M,N}\left(y|\theta \right)$, where θ is the maximising parameter of the model with respect to mean and variance and x is the total of the estimated parameters with sample size ‘n’. Since MM is a probabilistic framework, it can model nonlinear relationships and identify latent structures within the data. Therefore, it provides a more nuanced representation of the underlying biological processes of the microarray data. However, MM requires more computational resources and is often sensitive to the choice of model parameters. Next, we discuss the Fast Fourier Transform (FFT), which is computationally efficient compared to the MM.

#### 2.2.2. Fast Fourier Transform for DimRe

Fast Fourier Transform (FFT) is a frequency-domain technique that is computationally efficient and can be used as a DimRe technique. The FFT algorithm is a technique for calculating the discrete Fourier transform of a sequence for a time domain signal. The data are transformed from the time domain to the frequency domain using FFT, allowing for the detection and extraction of prominent periodic patterns naturally present in MAGE profiles. In this way, the relationships inherent in MAGE data are uncovered, and the subtle biological nuances could be unveiled. As given in Cheong Hee Park et al. [41], it is possible to simplify FFT by separately considering odd and even terms and also considering the periodic terms from the DFT expression given as:(3)$$X(c)=\sum_{n=0}^{N-1}x\left(n\right){W_{N}}^{cn}$$
where
(4)$$W_{N}=e^{-j2\pi /N}$$
here N stands for number of FFT points, n = 1, 2, 3 … N, c = 1, 2, 3 … N.

Similarly, the Inverse DFT (IDFT) is given by:(5)$$x(n)=\frac{1}{N}\sum_{c=0}^{C-1}X\left(c\right){W_{N}}^{-cn}$$

Next is a statistical analysis of the feature-extracted data to understand the changes in the dataset after the adopted DimRe methods.

#### 2.2.3. Impact Analysis of DimRe Methods through Statistics

The Pearson Correlation Matrix (PCM) can be used to analyse microarray gene expression data after dimensionality reduction, as performed by Kim et al. [42]. The PCM provides insights into the relationships between gene expression profiles by calculating pairwise correlations between genes across samples. Thus, PCM can uncover key gene clusters or functionally related modules, facilitating the discovery of cancer biomarkers. In this work, we evaluate the dataset with PCM after DimRe using the correlation coefficient *p*. The PCM measures the linear relationship between two data, ranging from −1 to 1. Here, *p* = 1 represents a positive linear relationship, *p* = −1 represents a negative linear relationship, and if *p* = 0, there is no relationship. Figure 2 explores the Correlation of the FFT DimRe method for Adeno and Meso cancer classes, respectively. For the Adeno class, correlation values lie between 0.63 and 1.00. So, there is a strong positive linear relationship within the data of the Adeno class. The positive relationship implies that the data in this class move together in a positive direction, and an increase in one variable is associated with an increase in the other. So, the data are more internally consistent and cohesive.

For the Meso class, correlation values lie between −0.08 and 0.27. Correlation values closer to zero suggest a weak or no relationship; hence, it is internally consistent. In essence, for the Adeno case, consistent patterns and relationships are available, and the Meso has inconsistent and diverse patterns and relationships. Figure 3 examines the Correlation of the MM DimRe Method for Adeno and Meso cancer classes, respectively. For the Adeno class, correlation values lie between 0.30 and 0.93. So, once again, there is a positive linear relationship within the data of the Adeno class. For the Meso class, correlation values lie between −0.10 and 0.14. So, the data presents a weak positive correlation and a negative correlation. Once again, Meso data are internally consistent. Overall, in the case of both FFT and MM DimRe techniques, weaker correlations pose a challenge for the classification model because it may need to rely on nonlinear relationships or interactions between features to distinguish instances of the class accurately. Also, the high correlations in the Adeno class pose a risk of overfitting. So, further data processing techniques like the FS method must be employed to mitigate the impact of these highly consistent and inconsistent correlated features to improve the classifier model’s generalisation performance.

After DimRe, we assess whether the selected features provide meaningful information about the underlying patterns in MAGE data. Statistical analysis helps validate the effectiveness of DimRe techniques by examining the significance of the extracted features about the target variable or the problem at hand. This section analyses whether the outcomes of MAGE data after DimRe is related to statistical parameters such as the mean, variance, skewness, *t*-test, kurtosis, CCA, *p*-value, and Pearson correlation coefficient (PCC). Table 1 represents the statistical features analysis for MAGE data after DimRe. As mentioned in Table 1, the FFT-based method displays higher mean values and variance among the classes.

The MM method depicts low mean and variance parameters, indicating a class part of variables within the cancer classes. All three types of DimRe methods give positive skewness values and flat kurtosis values. PCC shows a good correlation in intra-class outputs. The *t*-test and *p*-value reveal no significant nature after DimRe of the MAGE data. The canonical correlation coefficient indicates the strength and direction of the linear association between the canonical variables. The value of 0.3852 and 0.3371 suggests a moderate positive linear association between the two sets of variables after FFT and MM DimRe. So, there is some degree of association between the variables in the first and second classes, suggesting shared patterns or information between the two classes. Moderate CCA values after feature extraction positively affect classification by providing relevant information for distinguishing between classes.

In the previous discussion, the correlation plot delivered the correlation of data within each cancer class across various subjects in the database. However, it is important to visualise the two cancer classes combined to visualise the distribution of the overall dataset. The violin plot is a data visualisation combining aspects of a box plot and a kernel density plot, providing insights into a dataset’s distribution and probability density. The comparison of Adeno and Meso cancer classes FFT and MM methods are performed using the violin plot in Figure 4. The width of each violin represents the frequency of data points at different values. The range of the violin represents the comprehensive view of the data distribution. In Figure 4a, the Adeno data is distributed from 0 to 17,500 and Meso data from 0 to 20,000. In Figure 4b, the Adeno data is distributed from −75 to 180 and Meso data from −210 to 830. All the observations from Table 1 are reflected in Figure 4. Overall, DimRe reveals unseen patterns and creates complex relations in the data distribution between the two cancer classes. In essence, the DimRe enhances the pre-classification step. But still, techniques like FS are essential and must be put forth to avoid overfitting and underfitting issues during classification. Following this, in the next section, the FS technique is discussed.

### 2.3. Feature Selection (FS) Techniques

FS is a crucial step in classifying lung cancer data from MAGE data, as it discovers relevant genes that contribute significantly to the classification task while removing irrelevant and redundant features. Several FS techniques exist, including filter, wrapper, and embedded methods prevailing in the literature. Filter Methods like correlation-based involving information gain and mutual information techniques are performed over MAGE data in Almugren et al. [43], which delivered an 85% to 90% accuracy on different datasets. The filter techniques may not capture complex interactions among features that contribute to the classification of MAGE data. Wrapper Methods like Recursive Feature Elimination (RFE) and forward selection with backward elimination are used as FS techniques in lung cancer MAGE data classification by Cai et al. [44] and Alhenawi et al. [45] with an 86.54% and 94% accuracy, respectively. Wrapper Methods are computationally expensive and prone to overfitting, as they optimise based on the specific classifier’s performance on the training data. LASSO FS for tumour classification using MAGE data was tested in Kang et al. [46] with a 96% accuracy. A Random Forest-based FS was performed by Dagnew et al. [47] for cancer classification from MAGE data, with a 94% accuracy. LASSO and Random Forest-based FS sometimes fail to figure out the intricacies of MAGE data due to the diverse and nonlinear nature of MAGE data. However, Cui et al. [48] proposed Dragonfly FS for MAGE data that delivered 97% accuracy on lung cancer datasets. Based on the above reports in the literature, this research employs the meta-heuristic DragonFly (DF) Optimisation technique for FS on MAGE data after DimRe.

The DF is an optimisation technique influenced by dragonflies’ static and dynamic behaviour. In the research by Majdi Mafarja et al. [49], the binary version of the DF algorithm approach is employed to solve FS problems. The static swarming is for feeding and the dynamic swarming is for migrating. Dragonflies make small groups for feeding and fly over a small area to hunt their prey. But for migration, they will form large groups and fly in one direction over a long distance. In static swarming, the movement is not in a single direction, but follows a back-and-forth movement. These are the exploration and exploitation phases of a meta-heuristic algorithm. As represented by Chnoor M. Rahman et al. [50], separation, alignment, cohesion, attraction to food and distraction from the enemy are the important features of the DF algorithm.

Separation is the mechanism for avoiding collision with neighbours.
(6)$$S_{i}=-\sum_{j=1}^{N}X-X_{j}$$

Here S_i_ is the i-th individual’s separation motion, X is the position of the current individual, X_j_ is the position of the j-th dragonfly and N is the total number of dragonflies in the swarm.

Alignment is the matching velocity with the neighbours.
(7)$$A_{i}=\frac{\sum_{j=1}^{N}V_{j}}{N}$$

Here, A_i_ is the i-th individual’s alignment motion, V is the velocity of the j-th dragonfly in the neighbourhood. Cohesion represents the tendency of a neighbouring group towards the centre.
(8)$$C_{i}=\frac{\sum_{j=1}^{N}X_{j}}{N}-X$$

Here, C_i_ is the i-th individual’s cohesion, X is the position of the current individual, X_j_ is the position of j-th dragonfly, and N is the total number of dragonflies in the swarm.

Attraction to food can be calculated as
(9)$$F_{i}=X^{+}-X$$

F_i_ is the attraction to food for i-th individual, X is the position of the current individual, and X^+^ is the position of the source of food.

Distraction from enemies is as follows:(10)$$E_{i}=X^{-}+X$$

E_i_ is the i-th individual’s distraction motion from the enemy, X is the position of the current individual and X^−^ is the position of enemy. DF algorithm uses two vectors in an optimisation problem, step vector and position vector. The step vector is as follows:(11)$${∆X}_{t+1}=\left(sS_{i}+aA_{i}+{cC}_{i}+fF_{i}+{eE}_{i}\right)+w∆X_{t}$$
where w is the inertia weight, t is the iteration number, s indicates the separation weight, S_i_ gives the separation of the i-th individual, a represents the alignment weight, A_i_ shows the alignment of i-th individual, c is the cohesion weight, C_i_ indicates the cohesion of the i-th individual, f is the food factor, F_i_ gives the food source of the i-th individual, e represents the enemy factor, E_i_ is the position of enemy of the i-th individual, after calculating step vector, position vector can be calculated as follows:(12)$$X_{t+1}=X_{t}+{∆X}_{t+1}$$
here, t is the current iteration. Figure 5 depicts the impact of MM and FFT DimRe methods with DF FS for Adeno and Meso carcinoma cancer classes through the Normal Probability Plot. In ideal probability plot cases, a straight diagonal line suggests normality, aiding in identifying outliers and assessing the quality of data preprocessing. However, there are departures from linearity which indicate non-normality and the presence of subpopulations. There are distinct clusters or deviations from linearity, indicating nonlinearity and divergence within the dataset, representing the underlying subtypes and biological variations between cancer classes.

### 2.4. Classification

The prime objective of the research is lung cancer classification from MAGE data. As discussed previously, we use five classification algorithms from various observations in reported research: NR, NB, DT, RF and SVM (RBF). The presence of distinct clusters and subpopulations makes these classifiers perform better for the lung cancer MAGE data. MAGE data often exhibits nonlinear relationships, where the expression levels of genes may interact in complex ways to determine the class label. NR classifiers can better capture complex decision boundaries in the data. They can flexibly model complex relationships, potentially improving classification accuracy. Almugren et al. [43] have utilised NB as one of the classification techniques for cancer classification from MAGE data. The Naive Bayesian Classifier is based on the probabilistic principle, specifically Bayes’ theorem. NB will calculate the probability based on feature values, and then the class label will be allocated with the highest probability. NB is better suited for huge datasets due to the computational efficiency of the classifier. Decision trees, especially when deep and complex, can model these nonlinear relationships effectively.

Peng et al. [51] classified different cancer types from MAGE data, including lung, breast, and colon tumours. In MAGE data, where the number of genes can be very high, the ability of DT to automatically select relevant features can be advantageous. Mohapatra P et al. [52] used the Random Forest to classify medical data, which consists of eight datasets for different cancer types like breast cancer, prostate cancer, colon tumours and leukaemia. Random Forests, which are ensembles of decision trees, can further enhance the performance of decision tree models. They reduce overfitting, increase accuracy, and estimate feature importance. Random Forests are popular for MAGE data classification because they handle noise and variability in MAGE data. Huynh et al. [53] analysed SVM as a classification technique to classify MAGE data. An SVM classifier deals with the curse of DimRe by obtaining a hyper-plane in high dimensional feature space. In most cases, SVM produces sparse solutions, which will reduce computational burden and thereby improve accuracy. 

#### 2.4.1. Nonlinear Regression

NR is a statistical method that works on linear/nonlinear data. One of the powerful tools for analysing the data is linear regression. But in real-time scenarios, researchers have to deal with the mathematical models whose results are related to nonlinear predictor variables, as mentioned by Martín, C.A et al. [30]. The Euclidean distance is primarily considered from the target of MAGE data with input data using the following equation as represented in Wenseng et al. [54],
(13)$$\sum d={\left|\left|T_{i}-X_{i}\right|\right|}^{2}$$$T_{i}$ indicates the data target, and $X_{i}$ represents the input data with index i. A cuboid expression representing a 3D space is used to project ‘d’ in the following way.

The projection to the 3D space is expressed using the following cuboid equation:

Minimise: a = n_1_ × d + n_2_^2^ × d^2^ + n_3_^3^ × d^3^(14)

$$\mathrm{Subject}\;\mathrm{to}:\;\left\{\begin{array}{c} n_{1}>n_{2}>n_{3}>0,n_{i}[0,1]\;for\;i\;=1,2\;and\;3\\ {\left(\frac{n_{1}-n_{2}}{2}\right)}^{2}<0.5\\ n_{2}=\frac{n_{1}}{10},\;n_{3}=\frac{n_{2}}{10} \end{array}\right\}$$
later, f = min (a) is calculated and threshold function ‘s’ is chosen for the NR with b_0_ as the sum of the squares of average deviation.

s = f + b_0_(15)

The computation of the b_0_ is performed using the least squares method.

#### 2.4.2. Naive Bayesian Classifier

NB is based on Bayesian statistical principles. It is a simple and efficient classifier for MAGE data, as described by A. Kelemen et al. [55]. The equations and expressions for Naïve Bayesian Classifier are as follows: Let y_i_ be the class label for i-th training instance.
(16)Py=nN
where P(y) is the prior probability of class y, n is the number of instances of class y and N is the total number of training instances. The likelihood of class y can be calculated as
(17)$$P\left(x|y\right)=P\left(x_{1},y\right)\;*\;P\;\left(x_{2},y\right)\;*\;\ldots \ldots \ldots \;*\;P(x_{n},y)$$

The posterior probability of each class given the observed features x is found using Bayes theorem:(18)$$P\left(y|x\right)=\left(P\left(x|y\right)\;*\;P\left(y\right)\right)/P(x)$$

By evaluating the posterior probability of each class, it is possible to make predictions for a new feature vector x.

#### 2.4.3. Decision Tree Classifier

DT can be used for both classification and regression problems. It is a tree-like structure which uses decision nodes for making decisions, and leaf nodes represent the output of those decisions. It will start from the root node and traverse to the leaf node to make new predictions. The class label was stored at the leaf node. A feature and a threshold are used to split the data into two subsets at each node. In each node, the class labels will be in a mixed way. Hastie et al. [56] stated that information gain and Gini impurity are the common impurity measures for measuring it.

The information gain can be calculated as follows:(19)$$Information\;gain\left(S,X\right)=Entropy\left(S\right)-\sum_{u\in values\left(X\right)}\frac{{|S}_{u}|}{S}\;*\;Entropy(S_{u})$$
here X is the feature, S is the set of instances in a node, and S_u_ is the subset of instances with the value u of feature X. Gini impurity in given as follows:(20)$$Gini\left(p\right)=1-\sum_{j=1}^{N}{\left(p_{j}\right)}^{2}$$
where p represents the proportion of instances of class j in the node and N is the total number of classes. The goal is to select the feature and threshold to reduce maximum impurity, called best split. The best split can be represented by
(21)$$Bestsplit\left(S\right)={argmax}_{a,t}Impurity\left(S\right)-\sum_{u\in values\left(X\right)}\frac{{|S}_{u}|}{S}\;*\;Impurity(S_{u})$$
here X is the feature, S is the set of instances in a node, S_u_ is the subset of instances with the value u of feature X.

#### 2.4.4. Random Forest

Random Forest classifier is an effective technique for both regression and classification problems. It includes a set of decision trees. It takes predictions from each decision tree, and the final prediction will be based on the majority votes of prediction. It helps to improve the accuracy of prediction for the dataset. Random Forest works based on the technique of bootstrap sampling. There is a decision tree for each bootstrap sample. As discussed in previous cases, decision trees select the best split using information gain or Gini impurity criteria. At each node in the decision tree, a random subset of features is chosen to decide the split. Each tree in the Random Forest will independently make predictions, and the final prediction is taken by aggregating the votes of all the trees. The expression for prediction using the majority vote in Random Forest described by James et al. [57] is
(22)$$x={argmax}_{a}\sum_{n=1}^{N}x_{i}=a$$
where x is the final prediction, N is the number of decision trees in the forest, x_i_ is the prediction of the i-th tree, and ‘a’ is the class label.

#### 2.4.5. SVM (RBF)

The support vector classifier with the Radial Basis Function is a powerful classifier for handling nonlinear decision boundaries, as explained by El Kafrawy et al. [58]. The RBF kernel maps the input data into a higher dimensional feature space. Vapnik [59] pioneered the SVM concept, which emphasises finding a decision boundary that maximally separates the data points of different classes by maintaining a maximum margin, thereby enhancing the classifier’s generalisation capability. The margin, representing the distance between the decision boundary and the closest support vector, is crucial in determining the classifier’s robustness and ability to classify unseen data accurately. The margin is given by:(23)$$K\left(a_{i},a_{j}\right)=e^{\left(-\gamma {\left|\left|a_{i}-a_{j}\right|\right|}^{2}\right)}$$
here a_i_ and a_j_ are FS vectors of i-th and j-th occurrences, and the width of the radial basis function is controlled using the parameter γ. The objective function of SVM RBF can be represented as:(24)$${minimize}_{v}\frac{1}{2}\sum_{i=1}^{N}\sum_{j=1}^{N}v_{i}v_{j}x_{i}x_{j}K\left(a_{i}a_{j}\right)-\sum_{i=1}^{N}v_{i}\;\begin{array}{c} \mathrm{With}\;\sum_{i=1}^{N}v_{i}x_{i}=0\\ 0\leq v_{i}\leq R\;for\;i=1,2,\ldots \ldots N \end{array}$$
here, R indicates the regularisation parameter, x_i_ is the class label of the i-th instance, v is the dual vector, a_i_ is the i-th instance, and N is the total of instances. The prediction for a new instance is made by computing the value of the decision function. The decision function is given by:(25)$$g\left(x\right)=sign(\sum_{i=1}^{N}v_{i}x_{i}K(a_{i},a)+b)$$
here $K\left(a_{i},a\right)$ is the radial basis kernel function, a is the new instance, and b is the bias.

### 2.5. Training and Testing

As the MAGE dataset is limited, the research used the K-fold cross-validation method for training. Xiong et al. [60] explained that the dataset will be distributed into k–equal-sized subsets in the K-fold cross-validation technique. Each fold should have an almost equal distribution of classes. Then, k iterations should be performed where each iteration uses a different fold as the validation set, and the remaining folds used for training. In each iteration, the model will be trained using the training set, and the results will be assessed by means of the validation set. The process will be repeated until all folds are taken from the validation set. Once the k-fold cross-validation process has been completed, it is possible to retrain the full dataset, and new predictions can be made on unseen data. In this paper, a 10-fold cross validation is performed. There are 1253 features per patient in this work after DimRe. Mean Square Error (MSE) is utilised for supervising the training methodology.
(26)$$MSE=\frac{1}{N}\sum_{j=1}^{N}{\left(O_{j}-T_{j}\right)}^{2}$$

Here, T_j_ is the target value at model j and O_j_ is the observed value at time j.

Table 2 shows the training and testing MSE of the classifiers with and without FS methods for both MM and FFT DimRe methods. The training MSE always lies between 10^−7^ and 10^−9^, while the testing MSE changes from 10^−5^ to 10^−8^. The maximum number of iterations for the training process is 2000. The Naïve Bayesian classifier without FS method settled at a minimum training and testing MSE of 1.56 × 10^−9^ and 2.93 × 10^−7^, respectively. A SVM (RBF) Classifier with a Mixture Model DimRe method and DF FS scores a minimum training and testing MSE of 1.96 × 10^−9^ and 5.18 × 10^−7^ correspondingly. As in the case of the FFT DimRe method and with DF FS, the Nonlinear Regression classifier attained a minimum training and testing MSE of 2.54 × 10^−9^ and 6.24 × 10^−8^, respectively. The parameters and their values selected for classification are furnished in Table 3.

## 3. Results and Discussion

In this research, diverse ML algorithms are assessed with the help of a confusion matrix as given in Table 4, which uses 90% of input features for training and 10% for testing. 

For lung cancer detection, based on the confusion matrix shown in Table 4, the clinical situations are defined as:

True Positive (TP): A patient is accurately recognised with an Adeno cancer.

True Negative (TN): A patient is accurately recognised with Meso cancer.

False Positive (FP): A patient is wrongly recognised with Adeno cancer when they have Meso cancer.

False Negative (FN): A patient is wrongly recognised with Meso cancer when they have Adeno Cancer.

Next is an analysis of the different parameter metrics such as Accuracy, F1 score, MCC, Error Rate, Youden Index, and Kappa which can be used for analysing the performance. The equations corresponding to the different performance metrics used for evaluating the classifier performance are represented in Table 5. The Accuracy metric is used to evaluate the overall correctness of the classifier’s predictions, which is crucial for ensuring the reliable identification of gene expression patterns associated with lung cancer. The F1 score balances precision and recall between the imbalanced Adeno and Meso class distributions. The F1 score is important in this research as it aids in accurately identifying genes relevant to disease classification for the imbalanced dataset. The Matthews Correlation Coefficient (MCC) provides a balanced measure of classifier performance by evaluating models in datasets with varying class distributions. The proportion of misclassified instances is marked with Error Rate. The Error Rate offers insights into the classifier’s performance in accurately distinguishing between different gene expression profiles, which is essential for minimising false discoveries in microarray data analysis. The Youden Index is used to quantify the classifier’s ability to identify true positives while minimising false positives correctly. The Kappa metric measures the agreement between observed and predicted classifications, showing the repeatability of the produced classification results.

Table 6 depicts the performance of the classifiers based on metrics such as Accuracy, Error Rate, F1 Score, MCC, Kappa and YI for Mixture Model and FFT DimRe techniques without FS. From Table 6, it is shown that the Naïve Bayesian Classifier with the FFT DimRe technique performed with a high accuracy of 88.950%, an F1 Score of 93.464% and with a low error rate of 11.050%. The Decision Tree Classifier with the FFT DimRe technique performed with a low accuracy of 54.144%, an F1 Score of 66.122% and with a high error rate of 39.779%.

Table 7 depicts the performance analysis of the classifiers for the Mixture Model and FFT DimRe techniques with DF FS. It is clear from Table 7 that the Decision Tree Classifier achieved a high accuracy of 91.160%, an F1 Score of 94.558%, and a low error rate of 8.840% for the mixture model DimRe method. The Random Forest classifier is placed at the lower edge with a low accuracy of 53.039%, a high Error Rate of 46.961% and an F1 Score of 65.021%. The comparison with Table 6 and Table 7 reveals that the accuracy of the NB classifier is reduced from 76.243% to 68.508%. The reduction in accuracy is because NB assumes independence between features, and DF FS has removed certain independent features that are not directly correlated with class labels. Conversely, with the application of DF FS, the performance of SVM (RBF) is improved from 59.669% to 91.160%. This improvement in classification accuracy is because DF FS reduced the dimensionality of MAGE data by selecting subsets with informative features. With these fewer features, the SVM RBF classifier can more effectively model the MAGE data and avoid overfitting. Moreover, DF FS has retained the most discriminative features that helped the SVM RBF classifier to establish clearer decision boundaries that separates data in the higher-dimensional feature space.

The above uncertainty of classifier performance observed with DF FS is improved by employing Adam and RanAdam hyper-parameter tuning methods. Adam and RanAdam are adaptive optimisation algorithms that can efficiently adjust each parameter’s learning rate associated with the classifier during the training phase. This adaptiveness helps in navigating the parameter space more effectively. Therefore, an accelerated convergence leads to better solutions within a few iterations.

### 3.1. Hyper-Parameter Tuning

The objective of hyper-parameter tuning is to optimise the hyper-parameters of ML models to improve the performance as described by Daud Muhajir et al. [61]. Hyperparameters are parameters not learned from the data but are set before training the model. They can control various aspects of the training process. There are different approaches to determine the best values, such as the Adaptive Moment Estimation method (Adam), Stochastic gradient method, Relative Randomness Function (RRF), Random Weights (RW) hyper-parameter updating and Grid Search (GS) method, as indicated by Elgeldawi et al. [62]. RanAdam is a new hyper-parameter tuning method used in this work to improve the accuracy of lung cancer classification.

#### 3.1.1. Adam Hyper-Parameter Tuning

Adam is one of the commonly used optimisation algorithms used in training. It is more effective in handling non-convex optimisation problems as mentioned by Sena et al. [38]. The key parameters used by Adam for tuning are learning rate, β_1_, β_2_, ∈, and decay rates. $L_{r}$ (learning rate) controls the step size during parameter updates in the Adam algorithm. β_1_ and β_2_ were used to control the exponential moving averages of the gradient and its square, respectively. ∈ is a small constant added to the denominator in the Adam update rule to prevent division by zero. Adam often has optional learning rate decay mechanisms. It is possible to prevent overfitting by combining Adam with L2 regularisation. The strength of regularisation can be controlled by tuning the weight decay coefficient. Another parameter which has an impact on the convergence of Adam is batch size. Larger batch sizes can provide more accurate gradient estimates, while smaller ones can introduce more noise, which might require a smaller learning rate. The number of training epochs can also be considered a hyper-parameter, as stated by Kaur S et al. [63]. Finding the optimal number of epochs for each specific task is necessary. After defining the hyper-parameter space, we must select a tuning strategy like grid search, random search, Bayesian optimisation, etc. In this work, accuracy is chosen as the performance metric to optimise classifier parameters. For each hyper-parameter set in our tuning strategy, a classifier model is trained using Adam on the training data and validated on the validation set as indicated by Masud et al. [64]. Table 8 indicates the optimal and initial values of hyper-parameter tuning with Adam for different classifiers. The hyper-parameters are updated according to the following equation: (27)$$w_{t+1}=w_{t}-\frac{L_{r}}{\in +\sqrt{\hat{S_{t}}}}*\hat{V_{t}}$$
(28)$$\hat{V_{t}}=\frac{v_{t}}{1-\beta_{1}^{t}}$$
(29)$$\hat{S_{t}}=\frac{s_{t}}{1-\beta_{2}^{t}}$$
(30)$$v_{t}=\beta_{1}*v_{t-1}+\left(1-\beta_{1}\right)*\frac{\partial L}{\partial w_{t}}$$
(31)$$s_{t}=\beta_{2}*s_{t-1}+\left(1-\beta_{2}\right)*{\left(\frac{\partial L}{\partial w_{t}}\right)}^{2}$$

In the above equations, $w_{t}$ and $w_{t+1}$ denote to past and new hyper-parameters; $\frac{\partial L_{r}}{\partial w_{t}}$ refers to the loss function which has to be minimised according to hyper-parameter w.
(32)$$\frac{\partial L}{\partial w_{tr}}=\frac{{ER}_{tr}}{w_{in}},\;if\;tr=1$$
(33)$$\frac{\partial L}{\partial w_{tr}}=\frac{{ER}_{tr}-{ER}_{tr-1}}{w_{tr}-w_{tr-1}},\;if\;tr>1$$

Here, Error Rate is indicated by ER with tr as the present iteration and tr−1 as the previous iteration. Algorithm 1 illustrates the execution of classifier with Adam method.
**Algorithm 1.** Adam Hyper-parameter TuningStep 1.Start AlgorithmStep 2.Initialise iteration counter, t = 0Step 3.Initialise and assign values to hyper-parameters β_1,_ β_2,_ ∈, $L_{r}$, $w_{t}$, $w_{t+1}$, ${v_{t},s}_{t}$Step 4.Initialise parameters (weights) for the chosen classifierStep 5.Define the loss function to be minimised.Step 6.For each iteration t:Step 7.Compute the gradient of the loss function with respect to the hyper-parameters, $\frac{\partial L_{r}}{\partial w_{t}}$Step 8.Update the exponential moving averages of the gradient and its square, $v_{t}$ and $s_{t}$ using Equations (30) and (31)Step 9.Compute bias-corrected estimates of the averages, $\hat{V_{t}}$ and $\hat{S_{t}}\;$ using Equations (28) and (29)Step 10.Update the parameters (weights) or the chosen classifierStep 11.Calculate ER for the current equationStep 12.If tr=1, compute the gradient of the loss function with respect to the hyper-parameter $w_{in}$Step 13.Else if tr>1, compute the gradient of the loss function with respect to the hyper-parameter $w_{tr}$Step 14.Update the hyper-parameter $w_{t+1}$Step 15.If t = ConvCritStep 16.Go to Step 19Step 17.ElseStep 18.Go to Step 7Step 19.End Algorithm

#### 3.1.2. RanAdam Hyper-Parameter Tuning

RanAdam is a hyper-parameter optimisation technique that efficiently searches for the best hyper-parameters for ML models. Randomised Search is particularly useful when the hyper-parameter search space is large, and the computational resources are limited. The RanAdam method is introduced to improve the classification performance of Adam further. The procedure can be divided into Adam and Controlled Randomisation (CR). The Adam part of the algorithm is the same as performed previously and is used without any changes in the RanAdam method. The CR procedure in RanAdam is responsible for improving performance over the Adam method. Algorithm 2 represents the way of implementing RanAdam. The ideal values for hyper-parameters with high precision can be identified using the nested CR procedure inside the Adam algorithm. In other words, the CR will explore optimal and highly precise hyper-parameters neighbouring the values Adam’s method gives in each iteration. The Controlled Randomisation approach uses randomisation with two control parameters, such as solution considering rate and solution adjusting rate. The optimal and initial values of hyper-parameters β_1,_ β_2,_ ∈, $L_{r}$, $w_{t}$, $w_{t+1}$, ${v_{t},s}_{t}$ are considered to be the same as that of the Adam method. Algorithm 2 illustrates the execution of classifier with RanAdam method.
**Algorithm 2.** RanAdam Hyper-parameter TuningStep 1.Start AlgorithmStep 2.Initialise iteration counter, t = 0Step 3.Initialise and assign values to hyper-parameters β_1_, β_2,_ ∈, $L_{r}$, $w_{t}$, $w_{t+1}$, ${v_{t},s}_{t}$Step 4.Initialise parameters (weights) for the chosen classifierStep 5.Define the loss function to be minimisedStep 6.For each iteration t:Step 7.Compute the gradient of the loss function with respect to the hyper-parameters, $\frac{\partial L_{r}}{\partial w_{t}}$Step 8.Update the exponential moving averages of the gradient and its square, $v_{t}$ and $s_{t}$ using Equations (30) and (31)Step 9.Compute bias-corrected estimates of the averages, $\hat{V_{t}}$ and $\hat{S_{t}}$ using Equations (28) and (29)Step 10.Update the parameters (weights) or the chosen classifierStep 11.Calculate ER for the current equationStep 12.If tr=1, compute the gradient of the loss function with respect to the hyper-parameter $w_{in}$Step 13.Else if >1, compute the gradient of the loss function with respect to the hyper-parameter $w_{tr}$.Step 14.Initialise random numbers for Rand1, Rand2, Rand3, Rand4 and specify bandwidthStep 15.if rand 1 < solution considering rateStep 16.${w^{\prime }}_{t+1}={w^{\prime }}_{t}$Step 17.End ifStep 18.if rand 2 < solution adjusting rateStep 19.${w^{\prime }}_{t+1}={w^{\prime }}_{t}$ * bandwidth * rand 3Step 20.End ifStep 21.If ${w^{\prime }}_{t+1}$ < Lower bound (LB)Step 22.${w^{\prime }}_{t+1}=LB$Step 23.End ifStep 24.if ${w^{\prime }}_{t+1}$ > Upper bound (UB)Step 25.${w^{\prime }}_{t+1}$ = UBStep 26.End ifStep 27.if ${w^{\prime }}_{t+1}$ < UBStep 28.${w^{\prime }}_{t+1}$ = LB + rand4 * bandwidthStep 29.End ifStep 30.If (ER = minimum ER)Step 31.Optimum weight, ${w^{\prime }}_{opt}$ = ${w^{\prime }}_{t+1}$Step 32.ElseStep 33.Go to Step 14Step 34.If t = ConvCritStep 35.Go to Step 38Step 36.ElseStep 37.Go to Step 7Step 38.End Algorithm

The RanAdam method employed in this research uses the following values: bandwidth = 0.0095, maximum number of iterations = 100 or ConvCrit MSE, whichever is met first, solution considering rate = 0.6, solution adjusting rate = 0.9, Rand 1, Rand 2, Rand 3 ∈ (0, 1) and Rand 4 ∈ (0, 0.1). Next is the analysis of training and testing accuracy with Adam hyper-parameter tuning for MM and DDT DimRe techniques with DF FS. 

Table 9 shows the Training and Testing Accuracy Analysis of Classifiers with Adam hyper-parameter tuning for the Mixture Model and FFT DimRe technique with DF FS. Random Forest classifier shows the highest test accuracy of 91.95%, and SVM (RBF) shows a 93.79% training accuracy for the FFT DimRe Method and with DF FS. For the Mixture Model DimRe Method and Dragonfly FS, SVM (RBF) shows the highest accuracy for both training and testing at 98.66% and 96.47%.

Table 10 shows the training and testing accuracy analysis of classifiers with RanAdam hyper-parameter tuning for the Mixture Model and FFT DimRe technique with DF FS. The SVM (RBF) classifier shows the top test accuracy of 98.86% and 99.41% of training accuracy as well for the FFT DimRe Method and with DF FS. For the Mixture Model DimRe Method and Dragonfly FS, the Naïve Bayesian classifier shows the highest accuracy for training and testing at 93.22% and 95.87%.

Table 11 depicts the performance analysis of the classifiers with Adam hyper-parameter tuning for the Mixture Model and FFT DimRe techniques with DF FS. It is identified from Table 10 that SVM (RBF) achieved a high accuracy of 94.475% and an F1 Score of 96.667% with an Error Rate of 5.525% for the mixture model DimRe method. The Random Forest classifier is placed at the higher edge with an accuracy of 88.950%, an Error Rate of 11.050% and an F1 Score of 93.243% for the FFT DimRe method. 

Table 12 shows the performance analysis of the classifiers with RanAdam hyper-parameter tuning for the Mixture Model and FFT DimRe techniques with DF FS. It is identified from Table 11 that SVM (RBF) achieved a high accuracy of 98.343% and an F1 Score of 98.997% with a low Error Rate of 1.657% for the FFT DimRe method. The Random Forest and Naïve Bayesian classifiers are placed at the higher edge with the same accuracy of 91.160%, with an Error Rate of 8.840% for the FFT DimRe method. The F1 Score is 94.667% for the Naïve Bayesian classifier and 94.702% for the Random Forest classifier.

The Radar plot is depicted in Figure 6, which compares the classification methodologies researched in this paper. The analysis uses ten selected subsets of the main MAGE data based on high variability. Four classification methods are compared: Classification without DF, Classification with DF, Classification with DF and Adam, and Classification with DF and RanAdam. The angular axis (X) represents the various classifiers, and the radial axis (Y) represents the ten selected data sets. The distance of each data point from the centre on its corresponding axis indicates the classifier’s accuracy. The classification technique with maximum accuracy is the data point farthest from the centre on the X and Y axis. The Radar plot indicates that the classification with DF and RanAdam is the best performer. Also, in the Radar plot, there are large differences in distances between data points on the same axis. This spread of data points indicates significant performance variations, suggesting that some methods are more sensitive to data and parameter changes.

Finally, Table 13 shows the improvement in the Accuracy of Classifiers with Adam and RanAdam hyper-parameter tuning for the Mixture Model and FFT DimRe technique with DF FS. The Random Forest classifier has the highest improvement in accuracy of 41.81% with the RanAdam Method. The SVM (RBF) classifier has the lowest accuracy improvement of 3.509% with Adam hyper-parameter tuning.

### 3.2. Computational Complexity (CC)

The classifiers are studied by evaluating the CC. The CC is identified according to input O (n) size. CC is less if it equals O (1). The CC will increase as the number of inputs, ‘n’, increases.

Table 14 depicts the CC for all the classifiers among different DimRe methods with and without FS techniques. It is reported from Table 14 that the Naïve Bayesian classifier is at the level of low CC. The SVM (Linear) and Naïve Bayesian classifiers attained moderate complexity for the EHO and Cuckoo search FS methods across the three DimRe techniques. The Least Square Linear regression DimRe method with EHO FS leads to high CC overhead of the classifiers. The Random Forest classifiers across the DimRe methods with and without FS techniques induced high CC, but the achieved accuracy of the classifier is at the lower edge. The SVM (Linear) and SVM (RBF) classifiers perform well with moderate CC.

Table 15 displays the comparison of research work reported in this paper with the previous works on the lung cancer detection from the microarray gene using binary classifiers.

Our research handles the problem of noise and outliers, which are significant in microarray gene expression data using the integrated approach of MM and FFT dimensionality reduction techniques with Dragonfly feature selection techniques. FFT brings out periodic patterns and clustered data as they extract frequency-related features. The FFT alone does not directly handle noise and outliers; rather, the integrated FFT and Dragonfly feature selection reduces the dimensionality and noise by selecting the most discriminating features from the MAGE data. In the case of the Mixture Model (MM), the dimensionality is reduced so that the data are considered as a combination of multiple probability distributions. MM-based dimensionality reduction can capture the underlying structure of gene expression patterns, with noise and outliers affects. MM handles the noise and outliers by considering them as components with lower probabilities, effectively down weighting their influence on the overall model. The application of Dragonfly over MM dimensionality reduction will further reduce the noise reduction and dimensionality to simplify the classification overhead.

## 4. Limitations

The conclusions of this research may be restricted to the specific population of Adeno and Meso cancer classes and may not apply to other populations. The techniques proposed in this work depend on MAGE data, which may involve complex and expensive procedures that are not practicable for routine clinical trials. The presence of outliers in the data have a big role in the accuracy and reliability of the classification results in this work. An outcome of this study is the establishment of a comprehensive database for mass screening and sequencing cancer genomes. By incorporating MAGE data and adopting the proposed classification techniques, this database allows the identification of patterns and trends in cancer genomes. Early stage detection and prediction are paramount to improving cancer patients’ survival rates.

## 5. Conclusions and Future Work

The early detection of lung cancer has a very important role in improving treatment, thereby increasing the survival rate. The MAGE data analysis of lung cancer is an effective technique for early detection. This research combines ML techniques with MAGE data analysis to enhance lung cancer data classification. FFT and MM are used as DimRe techniques and DF is employed as an FS technique. The classification was completed using five classifiers with hyper-parameter tuning that were compared and their performance was evaluated. The result shows that the SVM (RBF) classifier with the FFT DimRe method and DF FS achieved the highest accuracy of 98.86% with RanAdam hyper-parameter tuning. The future work planned for this research is to employ LASSO as a method for dimensionality reduction and use ML classifiers, CNN classifiers, DNN classifiers, and LSTM methods for lung cancer classification from MAGE data.

## Figures and Tables

**Figure 1 bioengineering-11-00314-f001:**
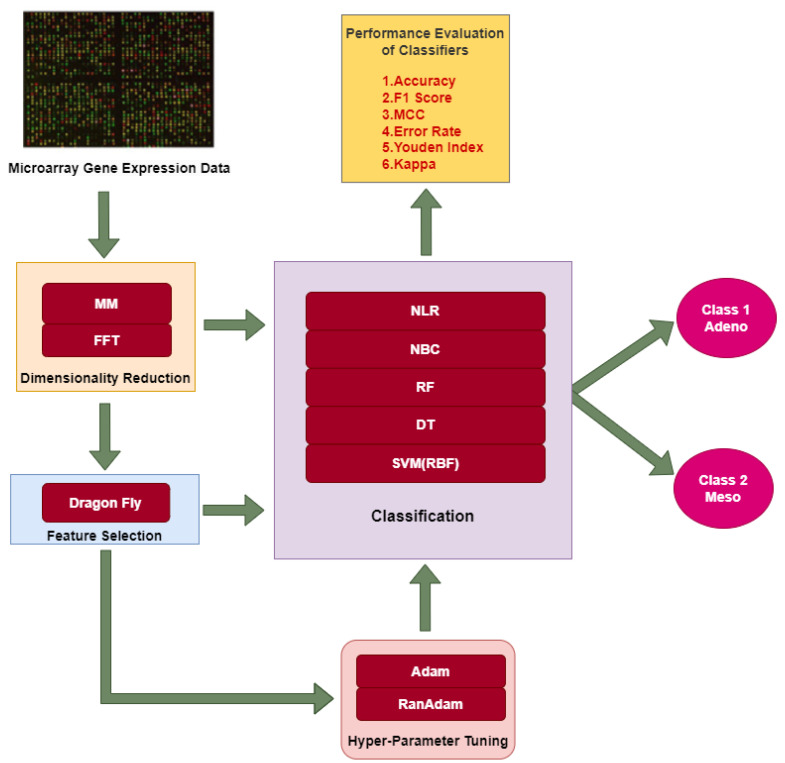
Different approaches employed in this research for classification of MAGE data.

**Figure 2 bioengineering-11-00314-f002:**
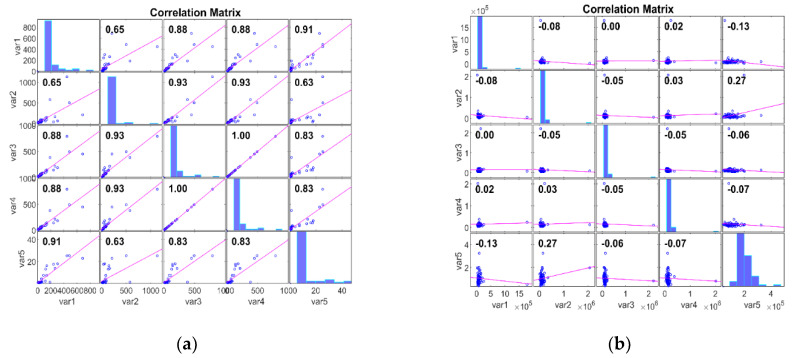
PCM plots for FFT DimRe methods. (**a**) PCM plot for FFT—Adeno class, (**b**) PCM plot for FFT—Meso class.

**Figure 3 bioengineering-11-00314-f003:**
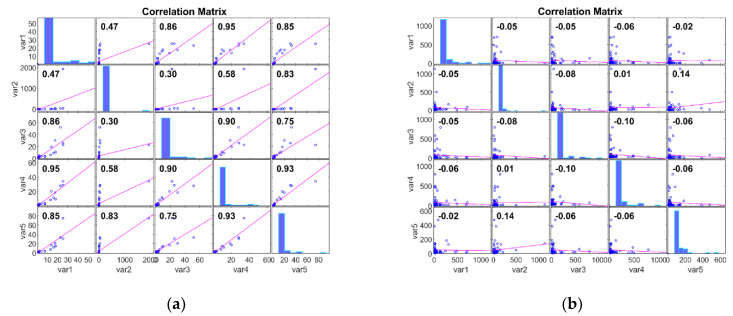
PCM plots for FFT DimRe methods. (**a**) PCM plot for MM—Adeno class, (**b**) PCM plot for MM—Meso class.

**Figure 4 bioengineering-11-00314-f004:**
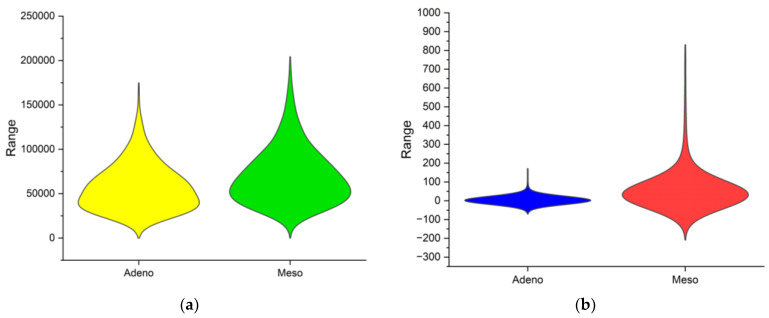
Violin plots of class distributions after DimRe. (**a**) Violin plot for MM method, (**b**) violin plot for FFT method.

**Figure 5 bioengineering-11-00314-f005:**
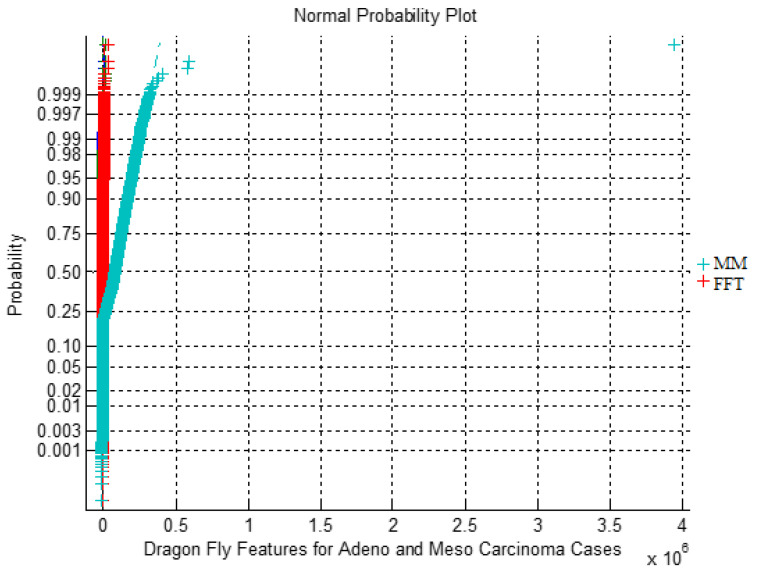
Normal Probability Plot for MM and FFT DimRe methods with DF FS for Adeno and Meso.

**Figure 6 bioengineering-11-00314-f006:**
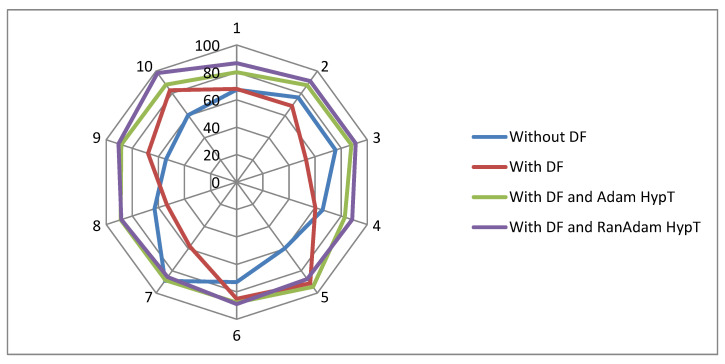
Radar plot for various classification methodologies employed in the paper.

**Table 1 bioengineering-11-00314-t001:** Average statistical features for mixture model and FFT dimensionally reduced Adenocarcinoma and Meso cancer cases.

Sl.No	Statistical Features	Mixture Model		FFT	
		Adeno Carcinoma	Meso Cancer	Adeno Carcinoma	Meso Cancer
1	Mean	12.77239	84.4254	50,051.74	64,399.1406
2	Variance	28,701.74	72,406.87	8.14 × 10^8^	1,207,801,420
3	Skewness	25.62594	11.83928	22.08858	17.9010876
4	Kurtosis	1008.477	211.3989	1392.65	1072.04601
5	PCC	0.84004	0.926835	0.944664	0.94001594
6	*t*-test	0.017655	3.14 × 10^−18^	2.06 × 10^−24^	1.096 × 10^−21^
7	*p*-value < 0.01	0.493103	0.5	0.5	0.5
8	Canonical Correlation Analysis (CCA)	0.3852		0.3371	

**Table 2 bioengineering-11-00314-t002:** Training and testing MSE of classifiers for Mixture Model and FFT DimRe technique without and with DF FS.

Classifiers	Mixture Model DimRe Method and without FS		FFT DimRe Method and without FS		Mixture Model DimRe Method and with DF FS		FFT DimRe Method and with DF FS	
	Training MSE	Testing MSE	Training MSE	Testing MSE	Training MSE	Testing MSE	Training MSE	Testing MSE
Nonlinear Regression	3.84 × 10^−7^	5.63 × 10^−5^	3.11 × 10^−6^	0.000016	1.44 × 10^−6^	3.6 × 10^−6^	2.54 × 10^−9^	6.24 × 10^−8^
Naïve Bayesian	1.56 × 10^−9^	2.93 × 10^−7^	5.61 × 10^−9^	3.24 × 10^−8^	3.48 × 10^−6^	4.2 × 10^−5^	3.03 × 10^−7^	5.04 × 10^−5^
Random Forest	1.23 × 10^−8^	1.94 × 10^−5^	1.44 × 10^−7^	6.89 × 10^−5^	3.06 × 10^−7^	5.76 × 10^−6^	6.4 × 10^−6^	2.92 × 10^−5^
Decision Tree	3.25 × 10^−6^	5.48 × 10^−5^	2.56 × 10^−6^	4.49 × 10^−5^	2.89 × 10^−7^	2.6 × 10^−5^	8.1 × 10^−7^	4.76 × 10^−5^
SVM(RBF)	2.6 × 10^−8^	1.69 × 10^−6^	8.1 × 10^−8^	2.5 × 10^−7^	1.96 × 10^−9^	5.18 × 10^−7^	1.02 × 10^−8^	1.56 × 10^−7^

**Table 3 bioengineering-11-00314-t003:** Classifier parameters and their values.

Classifier	Parameter Value
NR	T_1_ = 0.85, T_2_ = 0.65, n_1_, n_2_, and n_3_ is retrieved from (15), b_0_ = 0.01, Convergence Criteria (ConvCrit) = MSE
NB	Smoothing parameter, α = 0.06, Prior Probability = 0.15, ConvCrit = MSE
RF	Number of trees N_T_ = 100, Depth D = 10, ConvCrit = MSE
DT	Depth D = 10, ConvCrit = MSE
SVM (RBF)	Width of the radial basis function, γ = 1, ConvCrit = MSE

**Table 4 bioengineering-11-00314-t004:** Confusion matrix for binary classification.

Truth of Clinical Situation		Observed	
		Adeno	Meso
Actual	Adeno	TP	FN
	Meso	FP	TN

**Table 5 bioengineering-11-00314-t005:** Performance metrics for various classifiers.

Performance Metrics	Derived from Confusion Matrix
Accuracy	$Accuracy=\frac{\left(TN+TP\right)}{\left(TN+FN+TP+FP\right)}$
F1 Score	$F1=\frac{2*TP}{\left(2*TP+FP+FN\right)}$
Mathews Correlation Coefficient	$MCC\frac{\left(TN*TP-FP*FN\right)}{\sqrt{\left(\left(TP+FP\right)*\left(FP+TN\right)*\left(TN+FN\right)\right)}}$
Error Rate	$ErR=\frac{\left(FP+FN\right)}{\left(TP+TN+FP+FN\right)}$
Youden Index	$YI(\%)=\frac{TP}{TP+FN}+\frac{TN}{TN+FP}-100$
Kappa	$Kappa=(\frac{TP+TN}{100}-Eacc$)/(1-Eacc) Eacc=(((FP+TP)/100)∗(FN+TP)/100+(((FP+TN)/100)∗((TN+FN)/100))

**Table 6 bioengineering-11-00314-t006:** Performance analysis of classifiers for Mixture Model and FFT DimRe techniques without FS.

DimRe Method	Mixture Model					FFT				
Classifiers	NR	NB	RF	DT	SVM (RBF)	NR	NB	RF	DT	SVM (RBF)
Parameters										
Accuracy	67.403	76.243	75.691	65.746	59.669	72.928	88.950	62.983	54.144	60.221
F1 Score	78.067	84.912	84.397	76.692	70.445	81.369	93.464	74.131	66.122	69.492
MCC	0.197	0.307	0.317	0.179	0.194	0.404	0.583	0.170	0.067	0.315
Error Rate	32.597	23.757	24.309	34.254	40.331	27.072	11.050	37.017	45.856	39.779
Youden Index	24.839	35.505	37.398	22.839	25.742	51.978	53.398	22.065	8.839	41.763
Kappa	0.178	0.298	0.304	0.159	0.153	0.353	0.578	0.145	0.052	0.230

**Table 7 bioengineering-11-00314-t007:** Performance analysis of classifiers for Mixture Model and FFT DimRe techniques with DF FS.

DimRe Method	Mixture Model					FFT				
Classifiers	Nonlinear Regression	Naïve Bayesian	Random Forest	Decision Tree	SVM (RBF)	Nonlinear Regression	Naïve Bayesian	Random Forest	Decision Tree	SVM (RBF)
Parameters										
Accuracy	67.956	68.508	53.039	60.221	91.160	85.083	58.011	53.591	67.956	82.873
F1 Score	77.863	78.967	65.021	71.875	94.558	90.970	68.333	65.854	78.519	88.889
MCC	0.277	0.209	0.057	0.124	0.715	0.481	0.217	0.042	0.203	0.554
Error Rate	32.044	31.492	46.961	39.779	8.840	14.917	41.989	46.409	32.044	17.127
Youden Index	35.742	26.172	7.505	16.172	76.538	48.731	28.860	5.613	25.505	66.538
Kappa	0.240	0.191	0.043	0.103	0.711	0.481	0.163	0.033	0.184	0.524

**Table 8 bioengineering-11-00314-t008:** Optimal and initial values of hyper-parameter tuning with Adam for different classifiers.

Classifiers	Optimal Values				Initial Values		
	β_1_	β_2_	∈	$L_{r}$	$w_{t}$	$v_{t}$	$s_{t}$
NR	0.5	0.5	0.2	0.28	0.42	0.1	0.15
NB	0.6	0.4	0.26	0.32	0.5	0.1	0.2
RF	0.45	0.55	0.38	0.4	0.38	0.1	0.25
DT	0.55	0.45	0.33	0.41	0.6	0.15	0.2
SVM(RBF)	0.35	0.65	0.32	0.45	0.5	0.1	0.2

**Table 9 bioengineering-11-00314-t009:** Training and testing accuracy analysis of classifiers with Adam hyper-parameter tuning for Mixture Model and FFT DimRe technique with DF FS.

Classifiers with Adam Hyper-Parameter Tuning	Mixture Model DimRe Method and with DF FS		FFT DimRe Method and with DF FS	
	Training Accuracy	Testing Accuracy	Training Accuracy	Testing Accuracy
Nonlinear Regression	90.31	88.23	91.34	89.84
Naïve Bayesian	91.23	89.29	92.56	90.39
Random Forest	92.97	91.84	93.47	91.95
Decision Tree	86.31	82.87	92.54	90.39
SVM (RBF)	98.66	96.47	93.79	90.84

**Table 10 bioengineering-11-00314-t010:** Training and testing accuracy analysis of classifiers with RanAdam hyper-parameter tuning for Mixture Model and FFT DimRe technique with DF FS.

Classifiers with RanAdam Hyper-parameter Tuning	Mixture Model DimRe Method and with DF FS		FFT DimRe Method and with DF FS	
	Training Accuracy	Testing Accuracy	Training Accuracy	Testing Accuracy
Nonlinear Regression	92.62	89.74	92.44	90.64
Naïve Bayesian	95.87	93.22	93.52	90.51
Random Forest	94.25	92.86	94.62	92.19
Decision Tree	92.37	90.219	95.61	93.53
SVM (RBF)	93.66	90.72	99.41	98.86

**Table 11 bioengineering-11-00314-t011:** Performance analysis of classifiers with Adam hyper-parameter tuning for Mixture Model and FFT DimRe techniques with DF FS.

DimRe Method	Mixture Model					FFT Method				
Classifiers	Nonlinear Regression	Naïve Bayesian	Random Forest	Decision Tree	SVM (RBF)	Nonlinear Regression	Naïve Bayesian	Random Forest	Decision Tree	SVM (RBF)
Parameters										
Accuracy	80.110	87.293	87.845	82.873	94.475	87.845	88.398	88.950	88.398	87.845
F1 Score	87.413	92.256	92.667	89.199	96.667	92.466	92.929	93.243	93.023	92.414
MCC	0.417	0.570	0.572	0.494	0.805	0.618	0.607	0.631	0.586	0.630
Error Rate	19.890	12.707	12.155	17.127	5.525	12.155	11.602	11.050	11.602	12.155
Youden Index	47.849	59.075	57.183	56.301	80.538	67.419	62.968	66.194	57.849	69.978
Kappa	0.406	0.569	0.572	0.483	0.805	0.612	0.606	0.630	0.586	0.620

**Table 12 bioengineering-11-00314-t012:** Performance analysis of classifiers with RanAdam hyper-parameter tuning for Mixture Model and FFT DimRe techniques with DF FS.

DimRe Method	Mixture Model					FFT Method				
Classifiers	Nonlinear Regression	Naïve Bayesian	Random Forest	Decision Tree	SVM (RBF)	Nonlinear Regression	Naïve Bayesian	Random Forest	Decision Tree	SVM (RBF)
Parameters										
Accuracy	86.740	91.160	91.160	88.398	87.293	88.950	85.635	88.398	90.608	98.343
F1 Score	91.892	94.667	94.702	93.023	92.256	93.289	91.216	93.069	94.352	98.997
MCC	0.557	0.689	0.681	0.586	0.570	0.621	0.520	0.576	0.665	0.943
Error Rate	13.260	8.840	8.840	11.602	12.707	11.050	14.365	11.602	9.392	1.657
Youden Index	58.409	68.860	66.301	57.849	59.075	63.634	54.516	55.290	65.634	95.441
Kappa	0.556	0.689	0.680	0.586	0.569	0.620	0.519	0.575	0.665	0.942

**Table 13 bioengineering-11-00314-t013:** Improvement in Accuracy of Classifiers with Adam and RanAdam hyper-parameter tuning for Mixture Model and FFT DimRe technique with DF FS.

Classifiers	Mixture Model DimRe Method and with DF FS		FFT DimRe Method and with DF FS	
	Accuracy Improvement by Adam Method (%)	Accuracy Improvement by RanAdam Method (%)	Accuracy Improvement by Adam Method (%)	Accuracy Improvement by RanAdam Method (%)
Nonlinear Regression	15.172	21.65	3.145	4.347
Naïve Bayesian	21.519	24.84	34.375	32.258
Random Forest	39.623	41.81	39.752	39.375
Decision Tree	27.333	31.875	23.125	25
SVM(RBF)	3.509	4.43	5.66	15.73

**Table 14 bioengineering-11-00314-t014:** CC of the classifiers for FFT DimRe method without and with FS methods and hyper-parameter tuning.

Classifiers	Without FS	With DF FS	With DF FS and Adam Tuning	With DF FS and RanAdam Tuning
Nonlinear Regression	O (2n^3^ log2n)	O (2n^6^ log 2n)	O (2n^6^ log 2n)	O (2n^4^ log2n)
Naïve Bayesian	O (2n^4^ log2n)	O (2n^7^ log 2n)	O (2n^7^ log 2n)	O (2n^5^log2n)
Random Forest	O (2n^3^ log2n)	O (2n^6^ log 2n)	O (2n^6^ log 2n)	O (2n^4^ log2n)
Decision Tree	O (2n^3^ log2n)	O (2n^6^ log 2n)	O (2n^6^ log 2n)	O (2n^4^ log2n)
SVM(RBF)	O (2n^2^ log4n)	O (2n^5^ log 4n)	O (2n^5^ log 4n)	O (2n^3^ log4n)

**Table 15 bioengineering-11-00314-t015:** Comparison of previous work.

S.No	Author (with Year)	Database	Classifier	Classes	Performance Accuracy in%
1	Azzawi (2015) [31]	National Library of Medicine and Kent Ridge Bio-medical Dataset	SVM, MLP, RBFN	Adenocarcinoma, Meso	91.39 91.72 89.82
2	Gordon (2002) [39]	Gordon MAGE Data	MAGE ratios	Adenocarcinoma, Meso	90
3	Fathi et al. (2021) [65]	Gordon MAGE Data	Decision Tree with feature fusion	Adenocarcinoma, Meso	85
4	Guan et al. (2009) [66]	Affymetrix Human GeneAtlas U95Av2 microarray dataset	SVM (RBF) with gene based feature	Adenocarcinoma, Meso	94
5	Gupta et al. (2022) [67]	TCGA dataset	Deep CNN	Adenocarcinoma, Meso	92
6	Mramor et al. (2007) [68]	Gordon MAGE Data	SVM, Naïve Bayes, KNN, Decision Tree	Adenocarcinoma, Meso	94.67 90.35 75.28 91.21
7	Lin Ke (2022) [69]	Gordon MAGE Data	DT—C4.5	Adenocarcinoma, Meso	93
8	Daniel Xia et al. (2020) [70]	Gordon MAGE Data	Minimalist Cancer Classifier	Adenocarcinoma, Meso	90.6
9	Morani et al.(2021) [71]	TCGA and GEO Dataset	Multivariate cox regression analysis	Adenocarcinoma, Meso	90
10	This Research	Gordon MAGE Data	RanAdam Hyper-parameter tuning for FFT DimRe techniques with DF FS and SVM (RBF) Classification	Adenocarcinoma, Meso	98.34

## Data Availability

The original data presented in the study are openly available as provided in the paper authored by Gordon [39] at https://pubmed.ncbi.nlm.nih.gov/12208747 (26 February 2024).

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
