# Peer review of "Enhancement of Classifier Performance with Adam and RanAdam Hyper-Parameter Tuning for Lung Cancer Detection from Microarray Data—In Pursuit of Precision"

_bioengineering, 2024, doi:10.3390/bioengineering11040314_

Round 1
Reviewer 1 Report
Comments and Suggestions for Authors
#The way missing data is handled and other preliminary data processing operations are carried out can have a big impact on the results of microarray analysis. These preparation procedures may be essential for reproducing the study or using the methods on fresh datasets, however the report does not go into depth about them or how they affected the study's findings.
# Because of biological variability and changes in experimental settings, microarray data might be noisy and contain outliers. The study uses dimensionality reduction techniques such as FFT and MM, however it does not specifically examine how these algorithms manage noise and outliers, which are important aspects influencing the precision and dependability of cancer classification.
# Any study involving patient data must address these issues because ethical considerations and data privacy are becoming more and more important in medical research. The research methodology's ethical issues and patient privacy protections are not discussed in the publication.
Comments on the Quality of English Language
Need to check for minor improvements.
Reviewer 2 Report
Comments and Suggestions for Authors
The article focuses on Lung Cancer Detection from Microarray Data with classical machine learning algorithms. In classification, feature selection tests were performed using Adam and RanAdam Hyper Parameter Settings.
Equation numberings are incorrect.
Not enough information has been given about the data set. Is there a data imbalance? State the adeno and meso numbers.
In my opinion, there is no need to explain the classification algorithms in detail.
